# Soil Erosion Monitoring in Quarry Restoration Using Drones

**Vicenç Carabassa** [1,2] **, Pau Montero** [1] **, Josep Maria Alcañiz** [1,2] **and Joan-Cristian Padró** [2,*]

[1] Ecological and Forestry Applications Research Centre (CREAF), E08193 Bellaterra, Spain; v.carabassa@creaf.uab.cat (V.C.); p.montero@creaf.uab.cat (P.M.); josemaria.alcaniz@uab.cat (J.M.A.)

[2] Departament de Geografia, Edifici B, Universitat Autònoma de Barcelona, E08193 Bellaterra, Spain

[*] Correspondence: joancristian.padro@uab.cat; Tel.: +34-935-814-343

**Abstract:** Mining is an essential activity that supports the provision of raw materials. However, the extraction process of mining has deep environmental impacts. For this reason, restoration actions are mandatory, and monitoring is a key step in ensuring the renaturalization of affected areas. Erosion processes are one of the main problems that affect restored areas in extractive activities due to the frequently steep slopes and the difficulty of revegetating the technosols constructed using mining debris. This research aims to develop a method for determining soil losses due to water erosion in mine-restored areas by using Geographic Information Systems (GIS) and Remote Sensing (RS) tools. For the study, images obtained using Unmanned Aerial Systems (UAS) in an open pit mine in the process of restoration are used, from which the Digital Elevation Model (DEM) of the current state of the slopes is obtained (0.10 m spatial resolution). With GIS techniques, ridges of the rills and gullies generated in the slopes are detected, whereby an estimation of a first DEM before the erosive process and a second DEM after the erosive process can be constructed. Each of these DEMs are evaluated individually in order to determine the height differences and estimate the volumetric loss. At the same time, the results are validated with the DEM derived from official mapping agencies' airborne Lidar data (1.00 m spatial resolution), which yield consistent data in the volumetric quantification of the erosion despite the difference in spatial resolution. In conclusion, the high spatial resolution of drone images facilitated a detailed monitoring of erosive processes, obtaining data from vast and inaccessible slopes that are usually immeasurable with traditional field techniques, and altogether improving the monitoring process of mine restoration.

**Keywords:** mine restoration; soil degradation; DEM; drones; GIS tools; Remote Sensing



## 1. Introduction

Mineral resources are tremendously important to economic development in terms of raw materials and energy [1,2]. However, mining can cause severe damage to the landscape, soil and water, especially in extraction areas. Sustainable mining is a complete process that occurs during the lifespan of a mine, including the restoration of the affected land. In the restoration process, it is particularly crucial to monitor geotechnical risks, slope instabilities and soil losses, where action occurs in order to control key aspects, such as slope erosion [3]. Among the various types of soil erosion, gully erosion represents the severest and has significant on-site and off-site effects [4]. In a broad sense, gully erosion can be defined as an erosion process in which deep channels are generated by runoff water that remove topsoil to a certain depth [5]. In a more detailed sense, gully erosion is a threshold-dependent process that is controlled by a set of geo-environmental factors [6].

The eroded terrain caused by water erosion is characterized by a convergent topography that is associated with micro valleys (gullies) and separated by a series of divergent ridges and drainage divisions (elevated surfaces). Therefore, the basic geomorphological structure of any river landscape can be described by a net of bottom valleys with stream channels and a net of ridges with drainage divisions [7]. In eroded slopes where a network of rills, gullies and ditches was generated, a very detailed spatial sampling is necessary to

quantify and model the erosive process. It is at this scale of study where Remote Sensing (RS)-based on images captured by Unmanned Aerial System (UAS) takes on full meaning. UAS-based RS has become more modular, miniaturized and intelligent in recent years. With its unique advantages, such as high spatial resolution, low cost, short revisit cycle, efficient acquisition and easy operation, it is widely employed in various fields [8–10]. In recent years, the development of scientific research based on UAS data has enjoyed great progress and success in its application, even in combination with other Remote Sensing platforms, such as satellites or manned aircrafts [11,12]. However, the use of UAS applied on mining environments remains in constant development [13,14].

With their ease of use as well as time and cost efficiency in acquiring information, UAS equipped with cameras and sensors are increasingly replacing traditional methods. Thus, as unmanned aerial vehicle technology has advanced by leaps and bounds in the past decade, today, its use is indispensable in mining work and research. In this context, the studies focus on generating precision topography, 3D reconstruction, mining prospecting, controlling the opening of excavation fronts, monitoring geological risks and environmental control.

The recent widespread availability of Digital Elevation Models (DEMs) with high spatial resolution, such as those derived from digital photogrammetry and laser altimetry (Lidar), has facilitated the mapping of morphometric changes in the terrain. The joining of classical photogrammetric methods and novel Structure from Motion (SfM) techniques from UAS imagery, now makes possible DEM generation in an affordable way, planning imagery acquisition with enough overlap to reconstruct 3D point structures and further interpolate them into grid DEMs [11–15]. Generating DEMs with photogrammetric techniques requires a large number of control points to ensure its quality, either on the ground or photointerpreted from a previously acquired image [16]. Surveying DEMs with centimeter resolution derived from drone images allows the detection of elevation changes due to high precision, facilitating the monitoring of soil erosion processes [17].

The objective of the work is to generate detailed cartography of topography variations from images obtained by UAS as well as determine the loss of soil on the slopes of the mining area due to erosion by using Geographic Information Systems (GIS) and Remote Sensing tools. For this purpose, a methodology was developed starting from a DEM of high spatial resolution, which is based on the detection of watershed lines that are formed in the erosive channels on the slopes. The novelty of the method resides in the estimation of a DEM before the erosive process based on the interpolation of the points located on the ridges, taking advantage of the ultra-high spatial resolution of drone data. With these altitudinal references, the DEM prior to erosion is modeled to estimate the volume of eroded soil.

## 2. Materials and Methods

### 2.1. Study Area

The area of interest in which the drone flight was carried out is the extractive activity on the Lázaro quarry, which is located in the municipality of El Vendrell (Tarragona, Spain). The exploitation has an area of 60 ha and a perimeter of 3.70 km (Figure 1). The quarry can be divided into two types of lithologies. The northern zone is formed by a mass of limestone, dolomitic and marly limestone, from the Lower Cretaceous and the southern zone by an outcrop of calcarenites belonging to the Miocene. Usually, tertiary materials do not present erosion problems. However, in cases where there is anthropic activity and clearing techniques, they can suffer erosion and caving phenomena on the slopes due to the water flow acting on it. The potency of material with commercial value is approximately 106 m of hard limestone rocks [18]. The study is focused on a subarea that is located northeast of the quarry and has an area of 23,189 m$^2$. This subarea, which was restored in the early 2010s, is characterized by the formation of steep slopes (>30°) with a predominance of the slope-berm model. The lower slopes, which were restored

using waste material from the quarry itself, show no change and are practically devoid of vegetation and strongly affected by erosive processes.

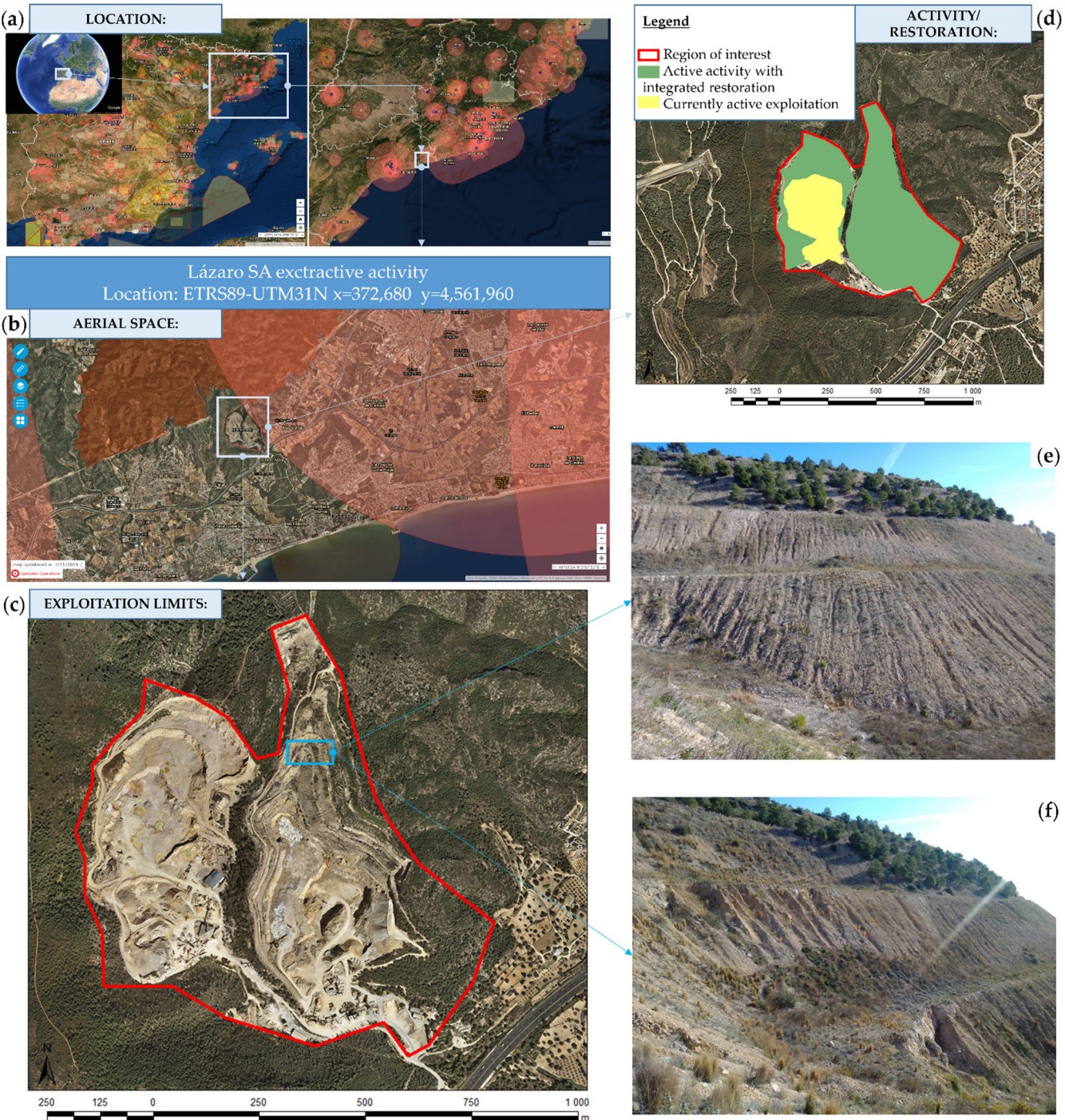

**Figure 1.** Location map of the limestone quarry studied. (**a**) General location of the study area in Catalonia, NE Iberian Peninsula. (**b**) Aerial space constrictions around the study area. (**c**) Exploitation limits and detail of the restored area. (**d**) Limits of the current extractive activity and limits of the restored areas. (**e**) Detail of the eroded slopes focused on the study. (**f**) Detail of the slope-berm restoration and erosion.

### 2.2. Unmanned Aerial Systems (UAS)

The unmanned aircraft used was an Atmos-7 [19] with flexible fixed wings. Having an autonomy of up to 70 min, the Atmos-7 allows a flight ceiling of 4000 m and a telemetry range of 5000 m. The sensor used for data collection was the MicaSense RedEdge-M camera, with a professional multispectral sensor capable of simultaneously capturing five discrete spectral bands to generate precise and quantitative information on the vegetation cover on the slopes (Figure 2). The flight was carried out on 8 January 2020, at a height of 120 m above ground, taking off and landing on an esplanade located a few kilometers from the quarry. With no clouds or wind, the good weather conditions allowed the flight to take place between 11:15 and 12:00 without repetitions (Figure 2).

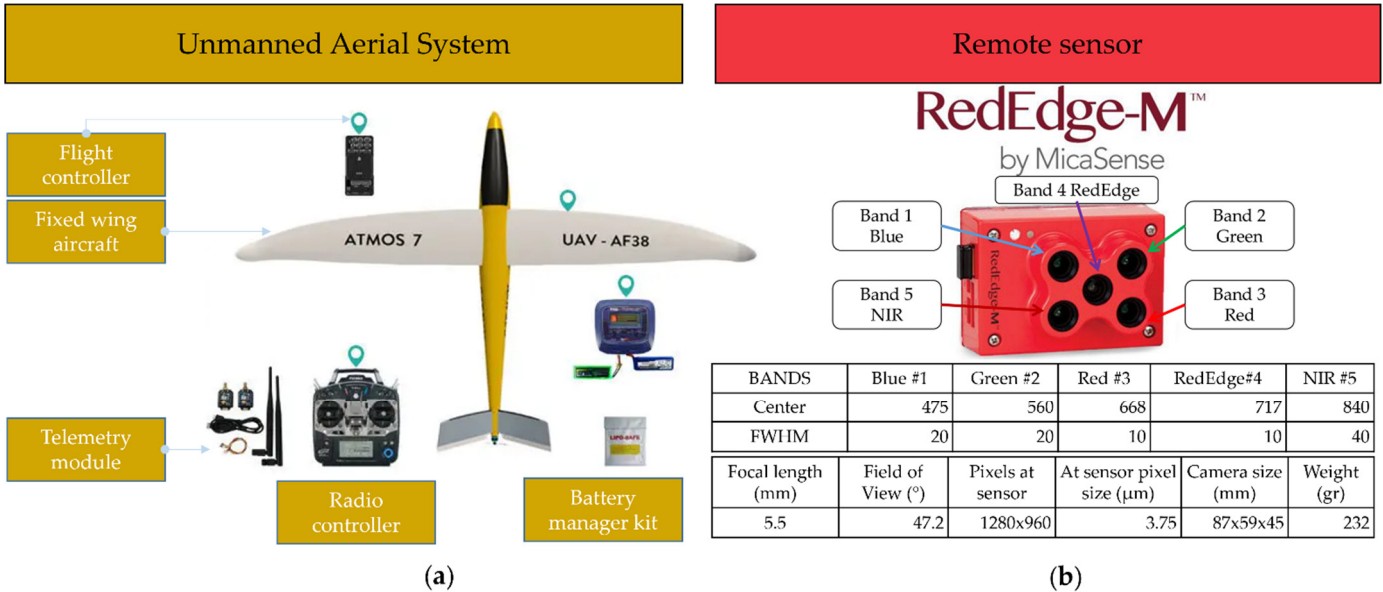

**Figure 2.** (**a**) Fixed-wing drone platform used for the mission. (**b**) Sensor used to capture aerial data.

From the overflight of the Lázaro quarry, a multispectral image and the DEM with a resolution of 0.10 m were obtained. Since the main purpose of the flight carried out was to co-register the resulting cartography with the official cartography, georeferencing was carried out by using 10 control points obtained by photointerpretation. The current orthophoto (0.25 m) of the Cartographic and Geologic Institute of Catalonia (ICGC) [20,21] for the planimetric coordinates (XY), and the DEM of 2.00 m × 2.00 m was also taken as a reference from the ICGC for the altimetric coordinate (Z). Three test points were used for achieving geometric accuracy in line with the reference pixel measurement (RMSEx = 0.11 m; RMSEy = 0.19 m; RMSEz = 0.45 m).

Therefore, with these materials, it is possible to obtain detailed enough imagery and digital models to accurately detect ridges and water troughs and manage them with GIS tools to obtain volumetric information of the eroded terrain.

### 2.3. Data Processing

The photogrammetric processing of the mosaic of the drone images, the georeferencing, the orthophoto generation, the 3D point cloud and the DEM was carried out with Structure from Motion (SfM) techniques using Agisoft Phostoscan photogrammetric software [11]. It is worth noting that the DEM was obtained using only the 3D points classified as ground in Agisoft Photoscan, a categorization performed in the SfM procedure thorough geometrical differential slopes and pixel radiometric features. From these data, three different ways of calculating the eroded area were applied: (1) relief inversion and drainage network detection, i.e., detection of ridges in the original elevation model; (2) the Flow Accumulation Model (FAM) [22]; and (3) the Relative Slope Position model (RSP) [23]. A

workflow consisting of twelve chained processes was applied (Figure 3). For the GIS and Remote Sensing processing, MiraMon Version 8.2 [24] and QGIS (SAGA GIS) [25] software were used for generating RSP and FAM models, which can be implemented using model builder tools (Figure 3).

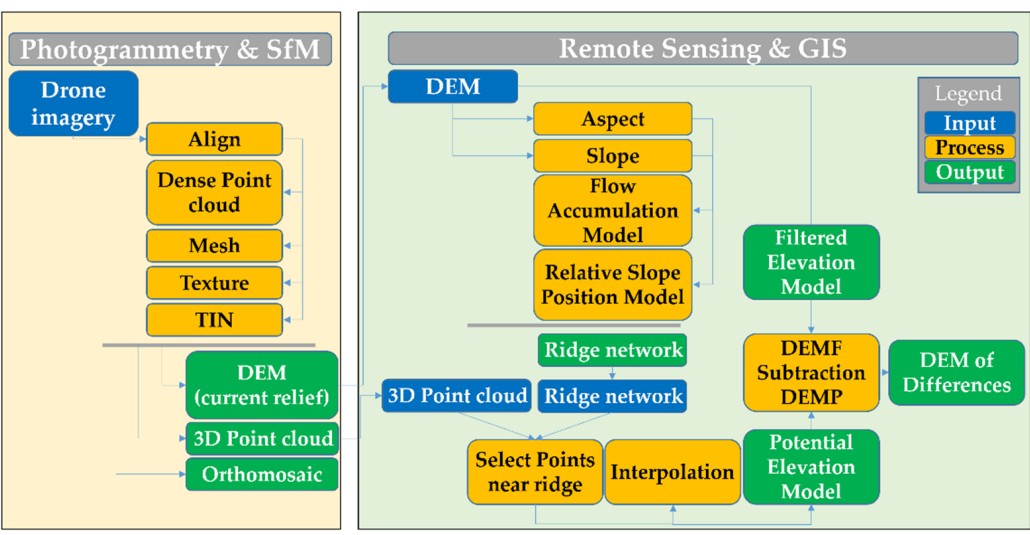

**Figure 3.** Comprehensive workflow of the described methodology, starting from the drone imagery and its Structure from Motion (SfM) processing, following with the obtention of the intermediate products (DEM of the current relief, 3D point cloud and orthomosaic), the GIS treatment of the DEM to obtain the ridge network, the estimation of the DEM before the erosive processes and the DEM of Differences (DoD) between the current relief DEM and the estimated original DEM.

The domains of the different drainage thresholds in the ridge pattern make it possible to delimit the accumulated runoff flow according to a well-defined criterion. With low thresholds, parallel ridges without intermediate cells occur with greater periodicity, whereas with high thresholds, they omit drainage dividers, and the threshold selection was such that it detected the largest number of ridges. Here, it should be noted that due to the complex nature of the surface, the process is only an approximation of the position of the ridges. In this complex and very detailed drainage network, orthophotography is used to check which scene-dependent threshold value is best to represent the specific landscape features.

To obtain the ridge network in vector format, the drainage order was defined, resulting in a raster that assigns a value of first order to the primary sections and second order to the secondary sections. The ridge mesh was then vectorized. To subsequently build a DEM without ruts or gullies, a buffer was generated around the ridges, and subsequently, a selection was made of the 3D point cloud obtained from the flight with respect to these polygons, thereby obtaining a new point cloud limited only to the points near the crest (without the points in the troughs of the slopes). To quantify the volume of rills and gullies, two DEMs estimated before erosion were generated, which were defined as the Potential Elevation Model (DEMP) and the Filtered Elevation Model (DEMF). The DEMP was modelled from the 3D points adjacent to the ridges using a Triangular Irregular Network (TIN) and Multilevel B-Spline [26], which subsequently transformed into a raster format resulting in a DEMP with the approximate height values as those before the erosion incisions. Apart from this, a DEM filtering of the current state of the slopes was performed with a median morphological operator. The median operator is a non-linear filter that is used to eliminate noise from images, whereby optimal use of this filter eliminates positive and negative valued spikes, being also an effective depression fill technique. Each pixel of the original raster is replaced in the destination raster by the outcome of the central tendency value, thus resulting in the DEMF. Finally, a Lidar point cloud was filtered, from



which the ground points were extracted and duplicate points were eliminated. Once the classified point cloud was obtained, the Elevation Model was created from the derived Lidar data (DEML).

The estimation of the height difference is a quantification of the value of the elevation changes within the slopes caused by erosion and deposition. It is calculated from the difference between the DEM of the current state of the ground and the DEM prior to erosion (Equation (1)).

$$DoD = DEM - DEM\ (p) \tag{1}$$

To calculate the volumes of gullies, the method applied was the subtraction of the DEM from the current state and the DEM prior to the erosive process. The volume calculation was carried out considering a global detection of the changes in the slopes, an uncertainty equal to the spatial resolution of the data (0.10 m) and a minimum detection level, which is obtained from the estimation of the error of the DEMs [27]. There are two different directions of volume change present in the study area: on the one hand, the continuous erosion leading to a negative volume change (material losses), and on the other, the positive values which represent interpolation artefacts and do not necessary correspond to a gain of material. This is because the methodology proposed in this work interpolates areas with concave terrain morphology where erosion has occurred. Therefore, convex areas of the relief (i.e., ridges between gullies, colluvial fan or other landforms) are detected as part of the relief unaffected by erosion processes.

The validation of such detailed data is difficult to carry out with field measurements. Therefore, we take as reference values for the erosion the DoD between airborne Lidar data obtained in 2010 and 2017 by the local official mapping agency (Institut Cartogràfic i Geològic de Catalunya, ICGC, Barcelona, Spain) [20,21]. Lidar data allowed the generation of two DEM of 1 m spatial resolution to quantify the volumetric differences in the 2010–2017 period.

## 3. Results

### 3.1. Gully Detection Methods

The automated detection of ridges using the proposed methodology for relief inversion is consistent with the visualization of the DEM and its derived products. As can be seen in Figure 4, buffer polygons derived from the drainage network of the inverted relief model are related to the areas of higher elevation along streams. These polygons represent the estimation of the topography before the erosion of the relief (Figure 4a).

In this study, the flow accumulation model has values ranging from 0.01 for the lowest accumulation pixels to 185 for highest accumulation pixels. A threshold of up to 5 was used to select DEM pixels that were not affected by the effects of erosion from surface runoff (Figure 4b). Choosing these parameters and limits is variable, depending on the morphological characteristics of the relief. The threshold determination is scene dependent and needs the human criteria to check with the ad-hoc orthophotography the ridges the algorithm performance.

In the RSP model, values range from 0 to 1. The value 0 represents the lowest relative height, while 1 represents the part with the highest relative topography. Pixels greater than 0.7 were selected, which represent the area of the relief that was not modified by erosion. Figure 4c shows the relative position of the relief.

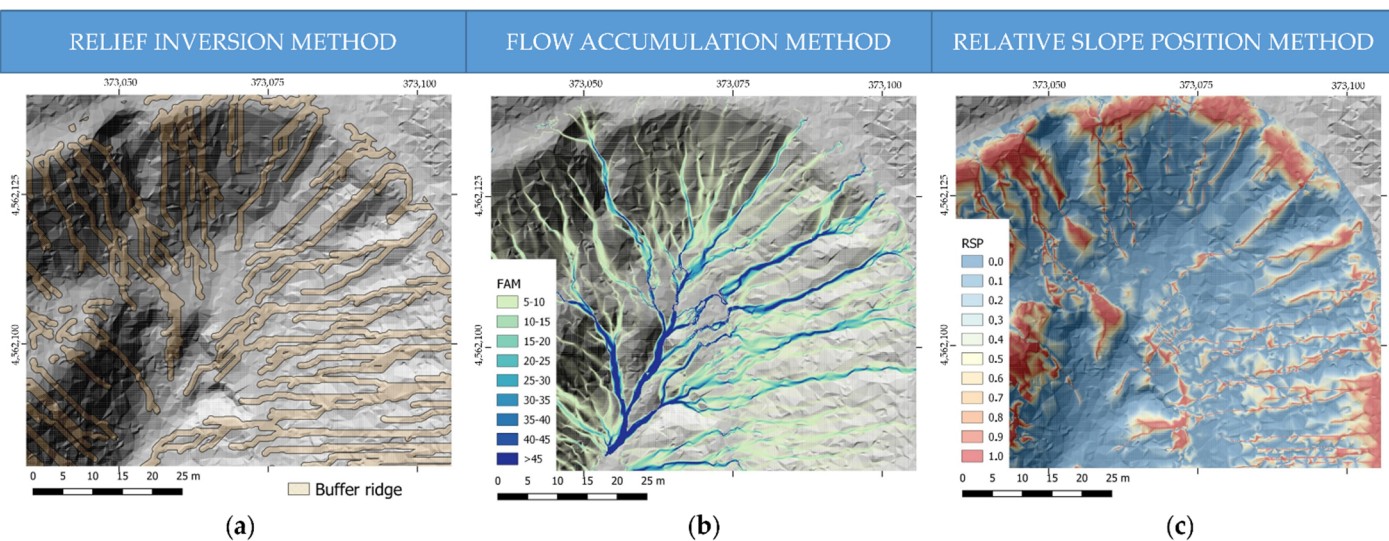

**Figure 4.** Detail of the cartographic representation of the different methods for detecting gully zones. (**a**) Inverse MDE method, in ochre, the buffer polygon. (**b**) Flow accumulation method, values range from 0 to 5 is represented in transparent which corresponds to the zones with a low flow accumulation or equivalent to the zone without erosion. (**c**) Relative slope position method, where values range from 0.7 to 1 represents areas not affected by erosion.

### 3.2. DEM of Differences (DoD)

The result of the difference between the models (DoD) is shown in Figure 5. The first case shows the interpolation of the pre-erosion DEM calculated by the relief inversion method (Figure 5a,d). The second case shows the interpolation of the model through the FAM (Figure 5b,e). Finally, the result of the model interpolated with the RSP is shown in Figure 5c,f. In these height difference models, the areas of greatest erosion are highlighted in color, whereby red indicates a correlation with the deepest gullies (Figure 5). The flow accumulation method shows the lowest erosion volume, while the model made with the relief inversion together with the RSP model interpolated by M-SPLINES shows the highest amount of eroded volume as well as eroded area (Table 1).

**Table 1.** Statistical data for the different gully detection methods used. The reference data for validation are the DEM of Differences (DoD) at 1.00 m spatial resolution obtained from airborne Lidar data sensed in 2010 and 2017 (local official mapping agency [20,21]).

| Raster Interpolation | Volume (m³) | Area (m²) | Min. Value (m) | Mean Value (m) | Stand. Dev. (m) |
|---|---|---|---|---|---|
| M-Spline Relative Slope Position | −5679.42 | 20,125.88 | −5.44 | −0.23 | 0.38 |
| TIN Relief Inversion | −5343.70 | 15,829.37 | −4.41 | −0.22 | 0.38 |
| M-Spline Relief Inversion | −5194.99 | 16,302.30 | −4.41 | −0.21 | 0.39 |
| TIN Relative Slope Position | −3342.82 | 13,594.35 | −5.71 | −0.13 | 0.29 |
| M-Spline Flow Accumulation Model | −1155.10 | 15,649.33 | −2.05 | −0.05 | 0.08 |
| TIN Flow Accumulatio Model | −541.96 | 7987.75 | −3.19 | −0.02 | 0.06 |
| Reference LIDAR 2010–2017 DoD | −2689.09 | 10,696.00 | −3.87 | −0.11 | 0.27 |

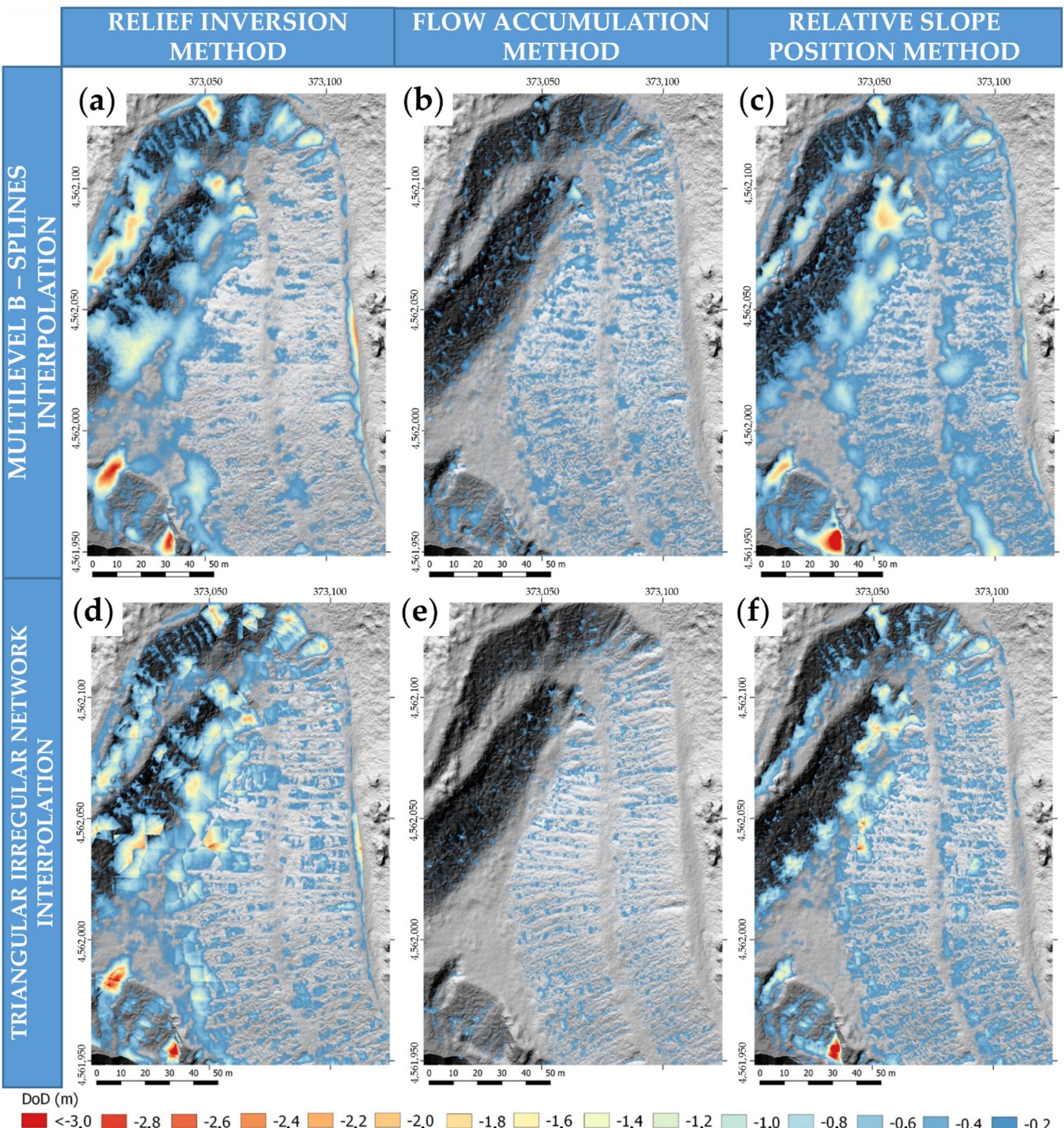

**Figure 5.** DEM of Differences (DoD) using different combination of relief modelling and interpolation methods: (**a**) through relief inversion and Splines interpolation; (**b**) flow accumulation and Splines interpolation; (**c**) relative slope position and Splines interpolation; (**d**) relief inversion and TIN interpolation; (**e**) flow accumulation and TIN interpolation; (**f**) relative slope position and TIN interpolation. The colors represent the depth of the gully. Values below 0.2 to 0 are shown in transparent.

## 4. Discussion

The results obtained in this study confirm the possibility of mapping erosion processes on slopes using different techniques for automatic gully detection. However, the quality of the results is strongly linked to the spatial resolution of the DEM used to perform the

work as well as the terrain control points used to adjust the DEM itself. Precise erosion estimation depends on the relative erosion rates on the slopes, and the mapping precision depends on the spatial resolution of the data as well as the vertical accuracy. So far, other studies have determined that a resolution of 0.05 m is sufficient for obtaining more than 85% of the information. However, this will ultimately depend on the study site and the geomorphology [2].

Among the models studied in this work, the flow accumulation model sufficiently detects the small gullies of the slopes, but it does not accurately detect the sectors where intense erosion occurred. On the other hand, by using both the RSP and the detection of crests by relief inversion, it is possible to detect the areas where greater erosion occurred, but the areas with more incipient erosion are not well detected.

This information could be complemented with the photo interpretation of the images for delimiting the gullies [28]. However, photointerpretation can be time-consuming and reduces the efficiency of the protocol, especially for monitoring large areas, such as those existing in extractive activities. Moreover, the use of Lidar data, if available, from rutinary airborne flights could also improve the proposed methodology [15]. However, the low point density of some rutinary flights (<2 point m$^{-2}$) make difficult the use this methodology, at least for detecting incipient erosion processes. Drone data give much more 3D point density, allowing us to work up to scales of 1:500 [12], although severely affected areas could be detected using Lidar data [21].

Vegetation creates problems when calculating the volume eroded on the land, as it generates artefacts that do not correspond to eroded areas. This problem could be partially solved with the exclusion of vegetation patches through vegetation classification using certain classification protocols [3,12].

It should also be noted that in the areas where the slope changes, i.e., from slope to plain, an artefact is produced. In models using RSP and inverted relief, such artefacts can be detected and measured. However, in flow accumulation models, such artefacts are not detectable (Figure 5). Interpolation by M-Splines tends to generate a greater difference in heights and occupy more area affected by erosive processes. However, with TIN interpolation, the surfaces are more conservative, and therefore, the difference between DEMs is usually smaller.

The comparison of the drone-acquired photogrammetric results with the airborne Lidar reference shows that the volumetric result of Relative Slope Position method is the most similar to airborne Lidar reference among the methods tested, but regarding the pixel minimum value, the Flow Accumulation and the Relief Inversion methods are closer to the reference. The validity and uncertainties of the results generated by this methodology should be evaluated in depth with field data surveys of the gullies on the slopes, so as to determine the success of the results in more detail. However, fieldwork can be difficult or even impossible in areas having steep and unstable slopes. For future research on debris erosion detection in slopes, one can envisage using higher spatial resolution together with control points on the ground in order to minimize the uncertainty in volume calculation.

## 5. Conclusions

In this study, a practical method to estimate the volume of eroded material in restored areas of an open pit mine is presented, using UAS imagery, photogrammetric and Remote Sensing techniques. Geographical Information Systems are the basis of the presented methodology, and we provide a workflow easily implementable in most GIS software thorough model builders. From the results, it is possible to obtain maximum and minimum estimations of the total volume on restored but eroded slopes and sedimentation, being especially relevant to identify where the main erosion processes take place. This approach facilitates the quantitative assessment of erosion processes, becoming a potential reliable tool for restoration programs, mining landscapes and many more slope erosion scenarios.

**Author Contributions:** Conceptualization, J.-C.P., J.M.A. and V.C.; methodology, J.-C.P. and P.M.; software, J.-C.P. and P.M.; validation, V.C., J.-C.P. and P.M.; formal analysis, V.C., J.-C.P. and P.M.; investigation, V.C., J.-C.P. and P.M.; resources, J.-C.P., J.M.A. and V.C.; data curation, J.-C.P. and P.M.; writing—original draft preparation, V.C., J.-C.P. and P.M.; writing—review and editing, V.C., J.-C.P., J.M.A. and P.M.; visualization, J.-C.P. and P.M.; supervision, J.-C.P., J.M.A. and V.C.; project administration, V.C. and J.M.A.; funding acquisition, V.C. and J.M.A. All authors have read and agreed to the published version of the manuscript.

**Funding:** This research was funded by the Government of Catalonia through the project "Research and Innovation in the process and restoration of extractive activities", by the European Union through the LIFE project "New Life for Drylands" (LIFE20 PRE/IT/000007), by Exodronics SL and by the Spanish Government through the Centro para el Desarrollo Tecnológico Industrial (CDTI) under the StratoTrans Project [IDI-20191270] (Proyecto cofinanciado por el Centro para el Desarrollo Tecnológico e Industrial (CDTI) en el marco de la convocatoria de Proyectos de I + D de Transferencia Tecnológica CERVERA).

**Acknowledgments:** The authors acknowledge the support of COMERCIAL LÁZARO SA.

**Conflicts of Interest:** The authors declare no conflict of interest. The funders had no role in the design of the study; in the collection, analyses or interpretation of data; in the writing of the manuscript, or in the decision to publish the results.

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
