# Peer review of "Soil Erosion Monitoring in Quarry Restoration Using Drones"

_minerals, doi:10.3390/min11090949_

Round 1

Reviewer 1 Report

Dear Authors,

The manuscript is interesting but needs some changes.

First of all, I feel like the aim of the paper must be clarified and must be in connection with the materials and methods and results presented.

I feel that the maps must be elaborated much more, including the visibility of the results to the legend.

The discussion part does not always seem to originate from the results.

E.g. "The flow accumulation model sufficiently detects the small gullies of the slopes,  . . ." - it is difficult to judge, maybe you can only say that it is the best for detection among the other methods used, especially that you stated yourself that: "The validity and uncertainties of the results generated by this methodology should be evaluated in depth with field data surveys of the gullies on the slopes, so as to determine the success of the results in more detail. However, fieldwork can be difficult or even impossible in areas having steep and unstable slopes."

You state that a methodology was developed but it is not clear which part of the results are considered by you as "a new methodology", as you presented several methods, however, you mention yourself that there was no field study, you also mentioned the scale but with no numbers mentioned, you also mentioned that LIDAR can help (for this there is a resolution mentioned in the abstract): "Moreover, the use of LIDAR data, if available, from rutinary airborne flights could also improve the proposed methodology", etc.

So, it is not clear what do you propose.

I think that the proposed changes are easy to fix, however, considering the magnitude of them for improving understanding, I propose major changes.

Regards, Reviewer X 

Reviewer 2 Report

Carabassa et al. present some methods for detecting erosion in a quarry from a low altitude overflight using a fixed-wing drone. They present a few workflows for detecting rills and gullies in topography, and the result from a few different types of digital terrain model (DTM as DEM should be reserved for models with a geoid referenced elevation). This comparison is interesting but not particularly novel, except perhaps as specifically applied to quarries and in the peer-reviewed academic literature. This is very much a technical report style of presentation rather than a scientific paper, and thus I am unsure if Minerals is an appropriate journal. There are several places in the manuscript where further explanation or detail is required. However, this article mainly suffers from a lack of a baseline for comparison. The authors are really kind of stuck without a conclusion because the values of Table 1 are simply presented without any sort of validation except against the other values. Thus, in Table 1, the reader is simply presented with 6 different models that produce very different results. Which is accurate, under what conditions, and is it any real improvement over other methods? How does this simple/easy model in this study compare to the more rigourous watershed modeling based on slope and runoff (e.g. Olivera & Maidment, WRR, 1999 among many other pubs)? Clearly there are some important background and experimental design issues to reconsider.

The figures in the paper are nicely produced and formatted. The figure captions are terse and need to be more descriptive of the data in each figure. Each caption is barely one sentence. Figure 3 colors make the text hard to read and only add small amount of information (via the legend). Paragraph in lines 54-61 gives a very sparsely referenced background on UAS data, which can be omitted. Paragraph in lines 69-76 conveys a naively simplistic view of the production of DTMs from aerial photogrammetry, which should be improved. Also line 129 should be changed from Surface from Motion to Structure from Motion. The methods section needs further explantion or specific information on the values and reasons for choosing those values on lines 141-2, 145, 148, 194, 195. The acronyms FAM and RSP are not defined in the text anywhere that I could find them. On line 150, what is a polyline of slope? On line 157, what is a ridge of a stream? Lines 160-163 suggest that a median filter is a depression fill technique, but instead a median filter is a despike filter and applies to both positive and negative valued spikes. 

Very concerning is that the multispectral data was not used to remove vegetation in the pre-processing steps. This may introduce significant error into the topographic models! Lines 246-249 are correct and the authors need to deal with this problem in this study. Finally, lines 257-262 should not need to be written because validation and uncertainty estimates are what are need to make this publishable.

While all of these are major revisions, I find the subject interesting and hope that the authors can improve their analysis.  

Reviewer 3 Report

  • The topic of the paper is interesting. Also, overall manuscript is well structured, the text is clear and, properly drafted. I conclude that this manuscript is suitable for publication after MINOR REVISION. Reviewers' suggestions may be of assistance to you in the preparation of revised papers.

  1. The authors used DEM data to monitor the topographical changes caused by soil erosion in the quarry. However, as can be seen from the right photo of Figure 1, there are large and small bushes on the surface. Although these bushes are small, there is a problem that causes errors in the surface height (DEM) and soil erosion amount. Also, if bushes are included in the drone photogrammetry technique, it can be considered that DSM data, not DEM, has been established. However, in this paper, there was no data processing or mention on the bush. The author needs to explain this in detail.

  1. [L79] Please use abbreviations (ex. remote sensing tools → RS tools) if you already state full term and abbreviations in earlier paragraph in the manuscript

  1. [L129] The authors state that “The photogrammetric processing of the mosaic of the drone images, the georeferenc-127 ing, the orthophoto generation, the 3D point cloud and the DEM was carried out with 128 Surface from Motion (SfM) techniques”. Please brief explanation on “Surface from Motion” in the manuscript. This is because, in general, the Structure from Motion (SfM) technique is used in photogrammetry data processing from UAV.

  1. [L131-132] I think it is a good choice to apply the difference of flow accumulation to show the change in DEM before/after soil erosion. However, the flow accumulation model mentioned by the author probably indicates the accumulated amount of rainwater moving in the direction of the maximum slope of the surface (without being absorbed by the surface). Therefore, although familiar to GIS majors, it will be more helpful to understand the concept by adding the term “surface runoff accumulation” to mine reclamation experts. Therefore, it is recommended to add the term surface runoff accumulation in parentheses when the term flow accumulation is first mentioned in the manuscript.

  1. [Figure 3] Most of texts in rectangle in Figure 3 is not legible at the present font style. Please check and revise the font style so that journal readers are able to decipher them.

  1. [L194-197] The authors state that “The flow accumulation model has values ranging from 0.01 for the lowest accumulation pixels to for highest accumulation pixels. A threshold of up to 5 was used to select DEM pixels that were not affected by the effects of erosion from surface runoff (Figure 4B). Choosing these parameters and limits is variable, depending on the morphological characteristics of the relief.” Flow accumulation greater than 5 were selected as unaffected area by erosion, on what basis? Is it the author’s subjective judgment or does the result of 5 from the FAM have scientific significance?

  1. [L199-292] The authors state that “In the RSP model, values range from 0 to 1. The value 0 represents the lowest relative height, while 1 represents the part with the highest relative topography. Pixels greater than 0.7 were selected, which represent the area of the relief that was not modified by erosion. Figure 4C shows the relative position of the relief.” Pixels greater than 0.7 were selected as unaffected area by erosion, on what basis? Is it the author’s subjective judgment or does the result of 0.7 from the RSP model have scientific significance?

  1. [Table 1] According to the result in Table 1, a large difference exists in the volume and area of soil erosion depending on the type of raster interpolation. The authors need to comment on this. For example, which technique is most appropriate for this case. This is because the text only mentions that TIN FAM shows the smallest value, but there are no other interpretation in the manuscript.

Round 2

Reviewer 1 Report

Dear Authors,

I see that the manuscript was improved based on the comments of all the reviewers.

I still need some reaction on some of my former, neglected comments, e.g. soil needs to be removed from the materials and methods, results and discussion and conclusion part as these gullies are formed by debris and there is no soil.

Also, the first sentence of the results is „The automated detection of ridges using the proposed methodology for relief inversion is consistent with the visualization of the digital shadow model.”, however, the digital shadow model shows up nowhere else in the manuscript. More explanation is needed!

Also, in the first paragraph of the result section, you write about pre-incision relief that does not show up anywhere else in the manuscript.

I do not list everything, please check all comments carefully and answer them, even if you think, it is irrelevant, let me know!

There are some minor issues, like if you refer to Figure A) in the text, it should not be Figure a) on the figure but the editors will deal with these issues, I assume.

I insist to include a description of methods mentioned only in the results (or later sections) earlier in the manuscript (Introduction or materials and methods). See my former comments on these issues.

I am looking forward to receive answers to my comments made in the former report.

Regards, Reviewer X

Reviewer 2 Report

The requested revisions were carried out in an adequate manner. The addition of more detail to the methods section is a great improvement. The addition of a table showing uncertainties in the method and some rudimentary consideration of the baseline data are much needed additions to this paper.
